# OpenReview forum: "WAInjectBench: Benchmarking Prompt Injection Detections for Web Agents"
_ICLR.cc/2026/Conference — ICLR 2026 Conference Withdrawn Submission_

### Official Review · Reviewer_nViD · 2025-10-15

**Soundness:** 2
**Presentation:** 3
**Contribution:** 2
**Rating:** 2
**Confidence:** 4

**Summary:**

The paper presents WAInjectBench, a comprehensive benchmark for evaluating prompt injection detection methods in web agent settings. It includes both text-based and image-based injection samples and evaluates the detectors using text and image modalities.

**Strengths:**

The writing is clear and easy to follow.

The experiment is comprehensive. It compiles a diverse dataset of injections from existing literature and studies 12 representative detectors (including prompting-, embedding-, and fine-tuning-based methods).

**Weaknesses:**

Overall, my main concern is that the paper does not provide additional insights into how to effectively protect web agents from prompt injection risks. In realistic deployments, users typically have no prior knowledge of whether injections exist, nor their modality (text or image), location, or injected content. This means we cannot know in advance which detector would be most effective. As a result, benchmarking detectors under pre-specified threat settings may have limited practical value.

Given this, it would be valuable to explore approaches like ensembling text- and image-based detectors, to assess whether such a unified detector yields robust detection performance when the nature of the threats is unknown.

In addition, many deployed web agents (e.g., Operator) already integrate built-in detection. It would be important to evaluate and compare with these commercial-level detectors. Agents also have self-evaluation mechanisms. These models can sometimes autonomously identify suspicious inputs during execution rather than relying on an external detector. Including such baselines would strengthen the study and clarify whether standalone detectors provide any unique advantages. This comparison could also offer practical guidance on whether future safe web agents should adopt independent detection modules or embed this capability directly within the model.

Another practical consideration is runtime efficiency. In real-world scenarios, detectors would need to operate in real time, potentially introducing non-trivial latency. Measuring and reporting the time cost of each detector would be an important complement to the accuracy-based evaluation. Furthermore, discussing when and how frequently the detector should be invoked would provide valuable insight. For instance, continuously running detection at every step may be infeasible—especially since many web-agent tasks (e.g., in OSWorld) require 100–200 interaction steps.


Question:

From my understanding, EIA involves injections with zero opacity, meaning the malicious instruction is not rendered visibly on the webpage. In that case, it is unclear how the image-based detectors achieved a TPR as high as 0.77. If the reported results correspond only to EIA samples with explicit instructions, this should be clearly stated in the paper, as it affects the interpretation of the experiment setup and conclusions.

**Questions:**

See Weakness

---

### Official Review · Reviewer_eAxj · 2025-11-01

**Soundness:** 3
**Presentation:** 3
**Contribution:** 2
**Rating:** 4
**Confidence:** 3

**Summary:**

The paper introduces WAInjectBench, a benchmark for detecting prompt-injection attacks against web agents. It formalizes a threat model, curates multi-modal datasets (text and images), instantiates families of detectors (prompting, embedding, fine-tuning, ensembling), and reports that detectors do fine when attacks are loud and explicit but crater on implicit or imperceptible perturbations.

**Strengths:**

1. Clear formulation of prompt-injection detection specifically for web agents rather than generic chatbots; grounded in the agent’s observation channels (screenshots, SoM/a11y/DOM/captions).

2. Curates multi-modal attack families (pop-ups, UI insertions, caption-based, WebInject, imperceptible VWA-Adv) with corresponding benign counterparts; reasonably broad coverage of real agent I/O.

3. Evaluates four detector families for both modalities (prompting, embeddings, fine-tuning, ensembles) rather than a single favored approach.

4. The paper shows a useful takeaway: detectors succeed on explicit instructions and visible artifacts, but fail on implicit or imperceptible perturbations; ensembling boosts recall at the cost of higher FPR.

**Weaknesses:**

1. Novelty: The claim of being the "first comprehensive" detection benchmark is under-argued relative to prior agent-security suites (e.g., WASP, VPI-Bench) that, while attack/eval-focused, will be viewed as adjacent.

2. Table 1 is not complete: Beyond malicious users and website owners, Table 1 should include a malicious platform participant [1,2], e.g., an Amazon seller who controls the content of their listing. This actor can inject instructions into product titles, descriptions, images, or metadata, yet is neither the end-user nor the platform owner.

3. Benign text distribution is mismatched: Negative text comes from ham emails/SMS on Kaggle and generated captions, which do not match web-UI distributions the agent actually observes; this likely understates FPR and miscalibrates thresholds for interface text.

4. Benign captions are generated with LLaVA-1.5-7B while LLaVA-1.5-7B is also a detector baseline (prompted and fine-tuned), risking optimistic FPRs via model-family coupling.

5. The strongest individual image baseline is GPT-4o-Prompt with low FPRs, making core results mainly rely on one proprietary model.

[1] Zhan, Q., Liang, Z., Ying, Z., & Kang, D. (2024, August). InjecAgent: Benchmarking Indirect Prompt Injections in Tool-Integrated Large Language Model Agents. In Findings of the Association for Computational Linguistics ACL 2024 (pp. 10471-10506).

[2] Zhang, J., Yang, S., & Li, B. UDora: A Unified Red Teaming Framework against LLM Agents by Dynamically Hijacking Their Own Reasoning. In Forty-second International Conference on Machine Learning.

**Questions:**

1. Decouple model families. I suggest decoupling the models used to generate captions from those used for detection, since sharing a model family can introduce bias.

2. Include more MLLMs. Incorporate additional multimodal LMs, both open and closed, for a more comprehensive assessment. Currently, there are few MLLMs evaluated.

3. Report efficiency metrics. Please provide per-sample latency, throughput, and cost for each detector family on common hardware to establish feasibility for real-time agent defenses.

4. From my perspective, we should replace or augment benign text with DOM-, a11y-, or SoM-derived interface strings and site-native emails or messages, so that false positive rates reflect the true web-agent distribution rather than artifacts like Kaggle ham/SMS datasets or LLaVA-generated captions.

---

### Official Review · Reviewer_BCyu · 2025-11-02

**Soundness:** 3
**Presentation:** 2
**Contribution:** 2
**Rating:** 4
**Confidence:** 3

**Summary:**

The paper develops a benchmark for prompt injection attack on web agents. The benchmark consists of six prompt injection attacks, six text-based and six image-based detectors, text dataset and image dataset.

**Strengths:**

This is a comprehensive benchmark on prompt injection threat eval for web agents. It includes advanced attack and defense strategies in a unified framework for fair comparisons.

**Weaknesses:**

1. It would be more convincing to validate why and how the LLM-based detector is optimal for this paper. Results with different models as backbones are helpful. Results with different prompts can be useful to show the optimality of the prompts the paper adopts and interesting findings may be drawn from it.

2. Using of TPR or FPR in main tables is questionable. Why not using F1 score?

3. There is no insight we can get from the evaluation here that motivates the community to build a more secure agent. As a benchmark paper, lack of useful insights/findings and motivation for future development will harm the contribution of this paper.

**Questions:**

Please refer to the weakness part.

---

### Official Review · Reviewer_bBa4 · 2025-11-02

**Soundness:** 3
**Presentation:** 3
**Contribution:** 2
**Rating:** 2
**Confidence:** 4

**Summary:**

This paper introduces WAInjectBench, a benchmark for evaluating prompt injection detection methods for web agents. It categorizes six attacks, builds text- and image-based datasets of malicious and benign samples, and benchmarks existing/self-built detectors (prompting, embedding, fine-tuning, and ensemble-based). The main finding is that detectors perform well on explicit attacks but fail on implicit or imperceptible ones.

**Strengths:**

1. Addresses an important and timely problem in web-agent safety.
2. Provides a comprehensive dataset across modalities.
3. Offers a systematic comparison of many existing detectors under one framework.

**Weaknesses:**

1. No evaluation of simple fine-tuning baselines.
2. In the proposed dataset, benign samples may not correspond directly to malicious ones. Including the benign version of the malicious ones could be helpful for furtuer comparison.
3. Limited novelty. The paper mostly aggregates prior attacks and detectors without new methods or deep analysis.
4. Shallow insights. The paper reports results but does not analyze failure cases or provide actionable design guidance.

**Questions:**

1. Do benign samples explicitly include benign versions of the malicious ones?
2. How would embedding and fine-tuning-based detectors perform when trained on subsets of your dataset?

---

### Note · Authors · 2025-11-12

I have read and agree with the venue's withdrawal policy on behalf of myself and my co-authors.